# Automatic Completion of Data Gaps Applied to a System of Water Pumps

**Ricardo Enguiça** *,†,‡ and **Filipa Soares** †,‡

Departamento de Matemática, Instituto Superior de Engenharia de Lisboa, Instituto Politécnico de Lisboa, Rua Conselheiro Emídio Navarro, 1, 1959-007 Lisboa, Portugal

\* Correspondence: ricardo.roque@isel.pt

† Current Address: CEMAT–CIÊNCIAS, Departamento Matemática, Faculdade de Ciências, Universidade de Lisboa, Campo Grande, Ed. C6, Piso 2, 1749-016 Lisboa, Portugal.

‡ These authors contributed equally to this work.

**Abstract:** We consider a time series with real data from a water lift station, equipped with three water pumps which are activated and deactivated depending on certain starting and halting thresholds. Given the water level and the number of active pumps, both read every 5 min, we aim to infer when each pump was activated or deactivated. To do so, we build an algorithm that sets a hierarchy of criteria based on the past and future of a given interval to identify which thresholds have been crossed during that interval. We then fill the gaps between the 5 min time steps, modeling the water level continuously with a piecewise linear function. This filling takes into account not only every water level reading and every previously identified change of status, but also the fact that activation and deactivation of a pump has no immediate effect on water level. This allows for the fulfillment of the ultimate objective of the problem in its real context, which is to provide the water management company an estimate of how long each pump has been working. Additionally, our estimates correct the errors contained in the time series regarding the number of active pumps.

**Keywords:** gap filling; time series; error correction; discrete-to-continuous model; water pump system

## 1. Introduction

A certain water lift station, receiving a continuous but variable and unknown flow of water, is equipped with three water pumps to release water and thus prevent the overflow of the tank. Activation and deactivation of the pumps depends on the water level (measured in meters), according to predefined thresholds, as given in Table 1 and explained next. The tank is also equipped with sensors that read, at 5 min intervals, the water level and the number of active pumps. In order to better understand the scope of our objectives, a small sample of the database, which spans an entire year, is displayed in Table 2.

We shall denote the thresholds 2.1, 2.3 and 2.5 using $U_1$, $U_2$ and $U_3$, respectively, and the thresholds 0.6, 1.2 and 1.9 using $L_1$, $L_2$ and $L_3$, respectively. We will refer to $U_1$, $U_2$ and $U_3$ as the *upward thresholds* and to $L_1$, $L_2$ and $L_3$ as the *downward thresholds*. We denote, using $s(n)$, the number of active pumps at time step $n$ and refer to it as the *status of the system* at time $n$.

The system operates according to the following rules. In case no pump is active and an increasing water level reaches $U_1$, the first pump is activated. Similarly, in case only one pump is (or, respectively, two pumps are) active and the water level reaches $U_2$ (or, respectively, $U_3$), a second (or, respectively, third) pump is activated. Analogously, in case all three pumps are working and the water level drops below $L_3$, one pump is deactivated;

if the level keeps lowering and reaches $L_2$, another pump stops working, and if the level reaches the level $L_1$, the remaining active pump is switched off. In other words:

- When none of the pumps is working, the status of the system will not change unless the water level reaches $U_1$;
- When only one pump is working, the status of the system will not change unless the water level reaches $U_2$ or drops below $L_1$;
- When two pumps are working, the status of the system will not change unless the water level reaches $U_3$ or drops below $L_2$;
- When all three pumps are working, the status of the system will not change unless the water level drops below $L_3$.

**Table 1.** Activation and deactivation thresholds.

| # of Active Pumps | Activation (m) | Deactivation (m) |
|---|---|---|
| 1 | 2.1 | 0.6 |
| 2 | 2.3 | 1.2 |
| 3 | 2.5 | 1.9 |

**Table 2.** Sequence of 20 rows of the original data set.

| Time | Reading | Water Level (m) | Number of Active Pumps |
|---|---|---|---|
| 12:10 | 0 | 0.699375 | 0 |
| 12:15 | 1 | 1.0975 | 0 |
| 12:20 | 2 | 2.1115625 | 1 |
| 12:25 | 3 | 2.338125 | 1 |
| 12:30 | 4 | 2.2928125 | 1 |
| 12:35 | 5 | 1.9346875 | 1 |
| 12:40 | 6 | 1.445 | 1 |
| 12:45 | 7 | 0.99875 | 1 |
| 12:50 | 8 | 1.2565625 | 1 |
| 12:55 | 9 | 1.4234375 | 1 |
| 13:00 | 10 | 1.2134375 | 1 |
| 13:05 | 11 | 0.8346875 | 0 |
| 13:10 | 12 | 0.7 | 0 |
| 13:15 | 13 | 0.9665625 | 2 |
| 13:20 | 14 | 2.51125 | 2 |
| 13:25 | 15 | 1.955625 | 1 |
| 13:30 | 16 | 1.6290625 | 1 |
| 13:35 | 17 | 1.2696875 | 1 |
| 13:40 | 18 | 1.576875 | 2 |
| 13:45 | 19 | 1.82 | 1 |

Figure 1 shows, for 60 consecutive entries of the database (the first 20 of which constitute the data given in Table 2), the water level and the corresponding number of active pumps as given in the data set.

Whereas the data relative to the water level can be considered accurate (apart from occasional typos, such as negative values and values significantly below $L_1$ or above $U_3$), the data relative to the number of active pumps is easily seen to contain errors. For example, in Reading 3 from Table 2, the correct number of active pumps must be at least 2, since the water level had already reached $U_2$, prompting a second pump to switch on (for a better visualization, we refer to Reading 4 on Figure 1, where the threshold $U_2$ is clearly attained). Another example is Reading 14, where the level $U_3$ was already reached, so the number of active pumps is 3, and it should only decrease to 2 at Reading 16, when $L_3$ is reached. Whenever detected, these values are replaced with one consistent with the remaining data, and we will make a thorough description in Section 2 of the criteria to determine the correct value at each time step.

In order to estimate for how long the pumps have been working, we model the level of the water continuously, so that we may estimate at which instant a pump has been activated or deactivated. To do so, we first identify under which conditions one or more thresholds have been attained in the interval of time under analysis, according to the data for the near past and near future of the interval. We then design an algorithm that models the water level, taking into account the switching on and off of the pumps, as well as an associated delay.

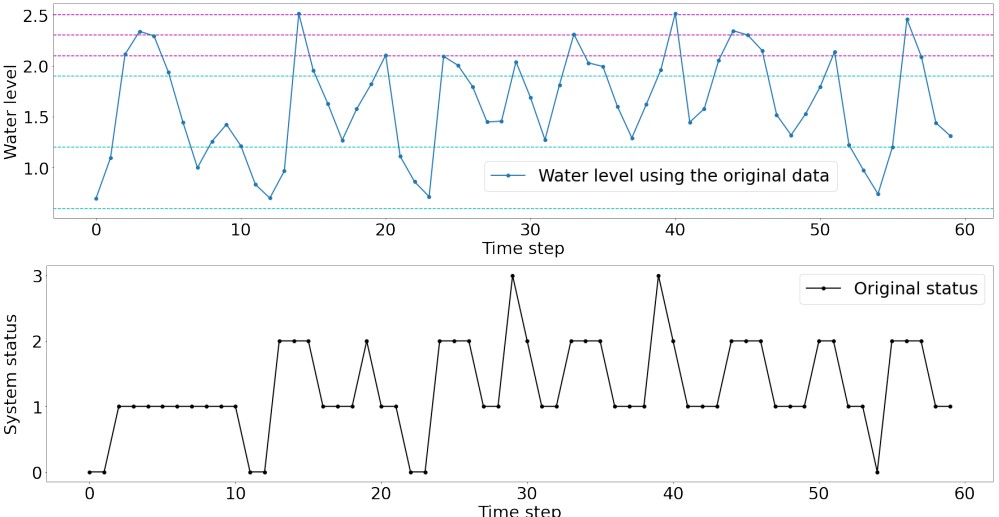

**Figure 1.** The water level and the number of active pumps for 60 consecutive entries of the database.

Given that both the data and common sense suggest that the effect of a pump being activated or deactivated may not be immediate and that the influx of water is variable and unknown, we consider a minimum delay of 5 s and a maximum delay of 15 s. Although empirical, these values are sustained by the data. For example, there exist readings above an upward threshold which are followed by a decrease in water level, meaning that the water level did not start to decrease immediately after the threshold was attained. Further evidence of this fact is the existence of values below 0.6 and above 2.5. Thus, let $d_m = 0.0167$ and $d_M = 0.05$ (measured in time steps). Delays within $d_m$ and $d_M$ are called *admissible*.

Concerning the literature on this framework, several recent works used machine learning tools to address problems, and goals, similar to ours. For example, in [1], the authors take an approach to fill gaps on a scarce data time series using the MissForest algorithm, and, in [2], gap filling is achieved by means of automated evolutionary identification of the optimal structure for a composite data-driven model. Our data set, however, contains a large number of errors in the status time series and thus does not provide a reliable training set for the application of machine learning regression techniques to infer the status of the system. Therefore, even though the status inference (with errors) from the water-level time series has an accuracy of 99% by means of the application of a random forest algorithm (cf. the public GitHub repository indicated at the end of this work), this will not allow for acquisition of the real status values, since it accurately predicts the status with those errors.

In [3], a multiple-point statistics algorithm suited for pattern reproduction is used. The main advantage of this method is its ability to provide probabilistic estimates of the missing values, which allows for uncertainty quantification. Although uncertainty is obviously also present in our problem, in our framework, pattern recognition benefits greatly from taking into account contextualized hands-on information about the system. For this reason, we favored an approach of a more deterministic nature, where uncertainty is addressed by providing criteria that establish which status change is most likely to have happened. The subsequent data gaps are then filled by means of several synchronized steps of linear interpolation.

A different, but related, possible approach to status inference would be an error-correction method, similar to those studied in [4] in an econometrics environment. Our framework is not related, since the corrections that we are interested in are based on empirical information and not on any difficulty in finding a fit model that performs the regression between the two time series.

Regarding data completion, we note that, in our problem, the gaps are the time values between the 5 min time steps instead of the erratic and larger gaps often considered in the literature. Several regression methods for gap filling between regular time steps can be found easily in the literature. For example, in [5], the author gives some insight on how to fit a discrete data time series into a continuous time function. Again, in our framework, the complexity arises from information outside the data set (threshold crossing and delays) and not with any difficulty in finding an appropriate regression model in particular. Other methods range from simple linear models to complex deterministic or stochastic techniques. Common approaches include, for example, the simple nearest neighbor method via data transfer [6], interpolation techniques [7], auto-regressive models [8], and simple and multiple regressions [9].

In short, to accomplish our goal, we first design an algorithm to predict the system status. Based on the previous and next water level readings of a given time step, this procedure identifies the thresholds, if any, that have been crossed between this and the following time step. Next, we define rules to model the water level, filling the gaps between the 5 min time steps continuously with piecewise linear functions. These rules follow these required principles:

- Take the water level readings from the original data set as constraints, since these are considered to be correct;
- Have the water level behave according to every previously identified change of status;
- Take into consideration the fact that the effect of the activation and deactivation of a pump on water level is not immediate, but rather comes with a certain delay.

Combining the status inference—that is, the estimated number of active pumps at each reading—with the estimated water level, we are able to determine the moment when each pump has been activated or deactivated. This allows for fulfillment of the company's ultimate objective, which is to estimate for how long each pump has been active.

This paper is organized in the following way: in the next section, we determine the conditions under which the status of the system changes; in the following section, we define how the gaps between readings should be filled for each type of transition; the fourth section is devoted to the implementation of Sections 2 and 3 on a sample of the data set; and in the final section, we present an analysis of the results obtained.

## 2. Inferring the System Status

As stated before, we consider the water-level time series to be accurate, and so any incoherence between the water-level readings and the number of active pumps is due to an error in the latter.

In this section we will describe a detailed procedure to obtain the correct status of the system at every time step $n$, and to do this we must consider what happens both before and after $n$. Moreover, the conclusions will be obtained using not only the water-level readings, but also the rate changes, in order to encompass the trend of the water level. The procedure not only addresses the reasons to infer changes in the status, but it also establishes prioritization criteria that allows us to settle which changes are the most likely to have occurred.

Let $r(n) = f(n) - f(n-1)$ denote the rate corresponding to the difference between the water level at time step $n$ and $n-1$. If the line that contains $(n-2, f(n-2))$ and $(n-1, f(n-1))$ and the line that contains $(n, f(n))$ and $(n+1, f(n+1))$ intersect within $[n-1, n]$, we denote the $x$-coordinate of this point using $x_I(n)$. Figure 2 exemplifies the existence and non-existence of $x_I(n)$.

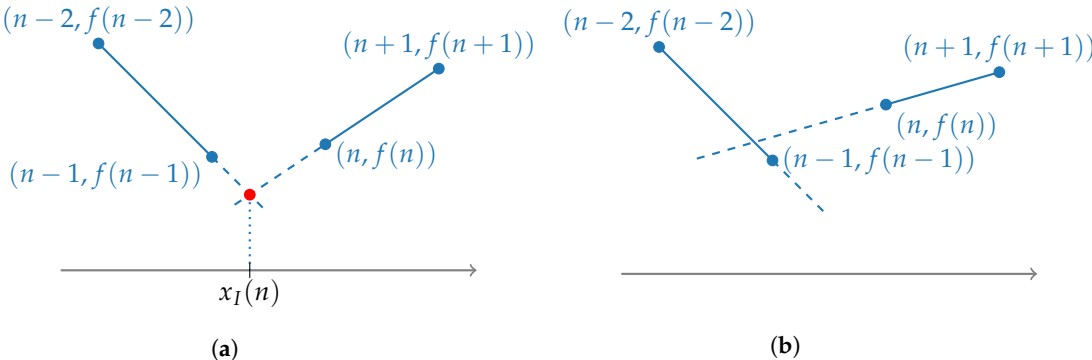

**Figure 2.** Example of existent and non-existent $x_I(n)$. (**a**) The $x$-coordinate of the intersection belongs to $[n-1, n]$. (**b**) The $x$-coordinate of the intersection does not belong to $[n-1, n]$.

An analysis of the data set allows us to infer some useful properties of the system, namely:

1.  $0.5 < f(n) < 2.6$ for the whole data set;
2.  $|r(n)| < 2$ for the whole data set;
3.  If $s(n) = 3$, then $r(n+1) < 0$ (this means that whenever all the pumps are simultaneously active, the water level decreases);
4.  If $s(n) = 0$, then $r(n+1) > 0.5$ (this means that the minimum inflow of water into the system is of 0.5 m per 5 min). As a consequence, since $U_1 - L_1 = 1.5$, the system is never fully turned off for more than three time steps;
5.  If $|r(n+1) - r(n-1)| > 1$ and $r(n-1) \cdot r(n+1) < 0$, then some threshold must have been crossed within $[n-2, n+1]$—if not within $[n-2, n-1]$ nor $[n, n+1]$, then within $[n-1, n]$;
6.  If $r(n) - r(n+1) > 0.6$, $r(n) \cdot r(n+1) < 0$ and $U_i - f(n) < 0.1$, then the upward threshold $U_i$ was crossed;
7.  Similarly, if $r(n+1) - r(n) > 0.6$, $r(n) \cdot r(n+1) < 0$ and $f(n) - L_i < 0.1$, then the downward threshold $L_i$ was crossed;
8.  If $U_i - f(x_I(n)) < 0.05$ or $f(x_I(n)) - L_i < 0.05$, then the corresponding threshold was crossed.

Next, we establish the conditions under which the system status changes from a time step $n-1$ to a time step $n$. For simplicity and ease of reference, these conditions are listed in Tables 3–6.

The main challenge is, of course, to identify which conditions imply that a certain status change takes place. Based on the analysis above, we consider four distinct conditions as evidence of a change from a status $k$ to a neighbouring status $k \pm 1$ and three distinct conditions as evidence of a change from a status $k$ to a non-neighbouring status $k \pm p$ with $p \geq 2$. Moreover, due to the fact that the transition from a status $k$ to a status $k \pm 1$ and the transition from a status $\widetilde{k}$ to a status $\widetilde{k} \pm 1$, both being transitions between neighbouring statuses, share similarities between them, the four conditions are precisely of the same type in both cases. The same applies to transitions between non-neighbouring statuses. Thus, overall, $s(n)$ will be different than $s(n-1)$ if one of the following occurs:

(a)  An explicit crossing of the relevant threshold;
(b)  Evidence of the trend before $n-1$ and the trend after $n$ meeting at a water level higher/lower than or close enough to the relevant threshold within $]n-1, n[$ more likely than later on;
(c)  For changes between neighbouring statuses, evidence of a sharp change from a positive/negative $r(n-1)$ to a negative/positive $r(n+1)$;
(d)  Evidence of a significant change from a positive/negative $r(n)$ to a negative/positive $r(n+1)$ coexisting in a reading close to the relevant threshold.

We shall detail the transition of $s(n-1) = 2$ to $s(n) = 3$, with the remaining transitions following analogous reasoning. As shorthand to make our arguments more concise, in the remainder of this work, both the conditions $f(x_I(n)) > f(x_I(n+1))$ and $f(x_I(n)) < f(x_I(n+1))$ will always be considered satisfied if $f(x_I(n))$ exists and $f(x_I(n+1))$ does not exist.

**Table 3.** Conditions for status change with $s(n-1) = 3$.

| $s(n)$ | Conditions |
|---|---|
| 0 | (30a) $f(n) \leq L_1$<br>(30b) $f(x_I(n)) < L_1 + 0.05$<br>(30d) $0 < f(n) - L_1 < 0.1, r(n+1) - r(n) > 0.6$ and $r(n) \cdot r(n+1) < 0$ |
| 1 | (31a) $L_1 < f(n) \leq L_2$<br>(31b) $L_1 + 0.05 \leq f(x_I(n)) < L_2 + 0.05$<br>(31d) $0 < f(n) - L_2 < 0.1, r(n+1) - r(n) > 0.6$ and $r(n) \cdot r(n+1) < 0$ |
| 2 | (32a) $L_2 < f(n) \leq L_3$<br>(32b) $L_2 + 0.05 \leq f(x_I(n)) < L_3 + 0.05$<br>(32c) $r(n+1) - r(n-1) > 1, r(n-1) \cdot r(n+1) < 0$ and $f(x_I(n)) < f(x_I(n+1))$<br>(32d) $0 < f(n) - L_3 < 0.1, r(n+1) - r(n) > 0.6$ and $r(n) \cdot r(n+1) < 0$ |

**Table 4.** Conditions for status change with $s(n-1) = 2$.

| $s(n)$ | Conditions |
|---|---|
| 0 | (20a) $f(n) \leq L_1$<br>(20b) $f(x_I(n)) < L_1 + 0.05$ and $f(x_I(n)) < f(x_I(n+1))$<br>(20d) $0 < f(n) - L_1 < 0.1, r(n+1) - r(n) > 0.6$ and $r(n) \cdot r(n+1) < 0$ |
| 1 | (21a) $L_1 < f(n) \leq L_2$<br>(21b) $L_1 + 0.05 \leq f(x_I(n)) < L_2 + 0.05$ and $f(x_I(n)) < f(x_I(n+1))$<br>(21c) $r(n+1) - r(n-1) > 1, r(n-1) \cdot r(n+1) < 0$ and $f(x_I(n)) < f(x_I(n+1))$<br>(21d) $0 < f(n) - L_2 < 0.1, r(n+1) - r(n) > 0.6$ and $r(n) \cdot r(n+1) < 0$ |
| 3 | (23a) $f(n) \geq U_3$<br>(23b) $f(x_I(n)) > U_3 - 0.05$ and $f(x_I(n)) > f(x_I(n+1))$<br>(23c) $r(n-1) - r(n+1) > 1, r(n-1) \cdot r(n+1) < 0$ and $f(x_I(n)) > f(x_I(n+1))$<br>(23d) $0 < U_3 - f(n) < 0.1, r(n) - r(n+1) > 0.6$ and $r(n) \cdot r(n+1) < 0$ |

**Table 5.** Conditions for status change with $s(n-1) = 1$.

| $s(n)$ | Conditions |
|---|---|
| 3 | (13a) $f(n) \geq U_3$<br>(13b) $f(x_I(n)) > U_3 - 0.05$ and $f(x_I(n)) > f(x_I(n+1))$<br>(13d) $0 < U_3 - f(n) < 0.1, r(n) - r(n+1) > 0.6$ and $r(n) \cdot r(n+1) < 0$ |
| 2 | (12a) $U_2 \leq f(n) < U_3$<br>(12b) $U_2 - 0.05 < f(x_I(n)) \leq U_3 - 0.05$ and $f(x_I(n)) > f(x_I(n+1))$<br>(12c) $r(n-1) - r(n+1) > 1, r(n-1) \cdot r(n+1) < 0$ and $f(x_I(n)) > f(x_I(n+1))$<br>(12d) $0 < U_2 - f(n) < 0.1, r(n) - r(n+1) > 0.6$ and $r(n) \cdot r(n+1) < 0$ |
| 0 | (10a) $f(n) \leq L_1$<br>(10b) $f(x_I(n)) < L_1 + 0.05$ and $f(x_I(n)) < f(x_I(n+1))$<br>(10c) $r(n+1) - r(n-1) > 1, r(n-1) \cdot r(n+1) < 0$ and $f(x_I(n)) < f(x_I(n+1))$<br>(10d) $0 < f(n) - L_1 < 0.1, r(n+1) - r(n) > 0.6$ and $r(n) \cdot r(n+1) < 0$ |

**Table 6.** Conditions for status change with $s(n-1) = 0$.

| $s(n)$ | Conditions |
|---|---|
| 3 | (03a) $f(n) \geq U_3$ <br> (03b) $f(x_I(n)) > U_3 - 0.05$ and $f(x_I(n)) > f(x_I(n+1))$ <br> (03d) $0 < U_3 - f(n) < 0.1$, $r(n) - r(n+1) > 0.6$ and $r(n) \cdot r(n+1) < 0$ |
| 2 | (02a) $U_2 \leq f(n) < U_3$ <br> (02b) $U_2 - 0.05 < f(x_I(n)) \leq U_3 - 0.05$ and $f(x_I(n)) > f(x_I(n+1))$ <br> (02d) $0 < U_2 - f(n) < 0.1$, $r(n) - r(n+1) > 0.6$ and $r(n) \cdot r(n+1) < 0$ |
| 1 | (01a) $U_1 \leq f(n) < U_2$ <br> (01b) $U_1 - 0.05 < f(x_I(n)) \leq U_2 - 0.05$ and $f(x_I(n)) > f(x_I(n+1))$ <br> (01c) $r(n-1) - r(n+1) > 1$, $r(n-1) \cdot r(n+1) < 0$ and $f(x_I(n)) > f(x_I(n+1))$ <br> (01d) $0 < U_1 - f(n) < 0.1$, $r(n) - r(n+1) > 0.6$ and $r(n) \cdot r(n+1) < 0$ |

For $s(n)$ to go from 2 to 3 between $n - 1$ and $n$, the water level must have reached the upward threshold $U_3$ during this interval. This is obviously the case if the water level is at least $U_3$ at $n$, that is, if $f(n) \geq U_3$. Since this is a condition of type "a", we shall denote it with (23a). Of course, it may also happen that, although $U_3$ was attained during the interval $]n - 1, n[$, the water level is no longer above $U_3$ by the time reading $n$ takes place. One evidence of this fact is that the trend before $n - 1$ and the trend after $n$ meet at a water level higher than or close enough to $U_3$ within $]n - 1, n[$, that is, $f(x_I(n)) > U_3 - 0.05$, along with an extra condition, $f(x_I(n)) > f(x_I(n+1))$, which ensures that the change is more likely to have happened in $[n - 1, n]$ than later on. Since this condition is of type "b", we labeled it (23b). Given that 2 and 3 are neighbouring statuses, a sharp change from a positive $r(n - 1)$ to a negative $r(n + 1)$ is also indicative of the water level having attained $U_3$ during $]n - 1, n[$. Thus, having $r(n - 1) - r(n + 1) > 1$ and $r(n - 1) \cdot r(n + 1) < 0$, along with the same extra condition, $f(x_I(n)) > f(x_I(n+1))$, which was included in (23b), is evidence of a status change from 2 to 3, and will be labeled (23c). Figure 3 illustrates conditions (23b) and (23c), according to whether $U_3 - 0.05$ sits below or above $f(x_I(n))$, respectively. Finally, if a significant change from a positive $r(n)$ to a negative $r(n + 1)$ coexists in a reading close to $U_3$, quantified as $0 < U_3 - f(n) < 0.1$, $r(n) - r(n + 1) > 0.6$, we also consider that the status changed from 2 to 3. This condition is labeled (23d). If none of the conditions (23a), (23b), (23c) or (23d) is met, we conclude that the status remains unchanged.

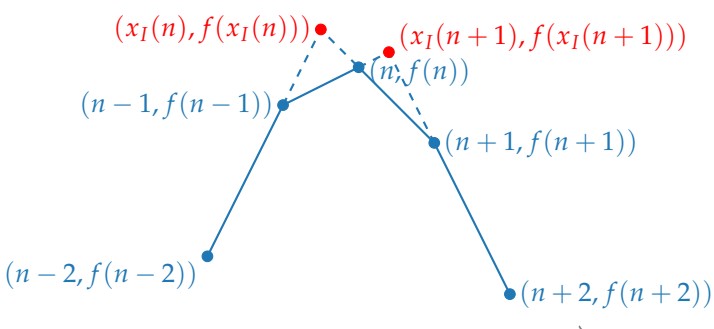

**Figure 3.** Illustration of conditions (23b) and (23c).

It is also important to note that the order in which these conditions should be verified, given a status $s(n - 1) = k$, is not entirely random. To prevent the change from being underestimated, the status farthest possible from $k$ must be checked first, the status second farthest from $k$ must be checked next, and so on (but it makes no difference whether one first checks if $s(n) = k - 1$ or $s(n) = k + 1$).

### 3. Data Gap Completion

Having established that the status of the system will change from a time step $n-1$ to a time step $n$—along with exactly which change will take place and the reason such a change will occur, that is, the type of condition ("a", "b", "c" or "d") that applies—we aim to continuously describe the water level, completing the information that the system does not convey regarding the water level. In this section, we establish how this completion should be accomplished for each of the status transition conditions identified in the previous section. We detail the procedure to follow for crossing the upward thresholds, since crossing the downward thresholds has entirely symmetrical consequences on the water level. Recall that we always allow for a delay, between $d_m$ and $d_M$, from the instant when a pump is activated or deactivated to the instant when its effect on the water level takes place.

To ease the notation, the water level at time step $n$ will be denoted as $f_n$ instead of $f(n)$, and the rate $r(n) = f(n) - f(n-1)$ will be denoted as $r_n$. If the line segment that joins $(n-1, f_{n-1})$ and $(n, f_n)$ intersects $y = U_i$ or, in case this intersection does not exist, if the line segment that joins $(n-1, f_{n-1})$ and $(x_I(n), f(x_I(n)))$ intersects $y = U_i$, we denote this point as $(x_{U_i}(n), U_i)$.

In case there is no change of status between readings $n-1$ and $n$, the water level is given by the linear function that joins $(n-1, f_{n-1})$ and $(n, f_n)$. The task is thus to establish how to fill the gaps when it has been determined that the status changed between two consecutive readings. The procedure must ensure that the relevant threshold(s) was(were) indeed crossed, verify the existence of a delay within the given limits, and verify a change in the rate of the water level once this delay is fulfilled. This is accomplished with a piecewise linear function, with the smallest possible difference from the linear function connecting the readings.

The procedure also takes into account the type of condition that sustains the status change. In fact, if the condition is of type "a", we have a reading above/below the relevant threshold, so that it is guaranteed that the water level will cross the threshold (and the adjustments to make, if any, have to do with ensuring an admissible delay), whereas in a condition of type "b", this crossing must be accomplished between $n-1$ and $n$. Likewise, conditions of type "c" and "d" have their own specificity. On the other hand, the procedure functions differently accordingly to the number of thresholds crossed. Therefore, we consider the following cases (recall that conditions of type "c" only exist for transitions between neighbouring statuses):

| | |
|---|---|
| A1/A2/A3: | for transitions of type "a" in which 1/2/3 thresholds are crossed; |
| B1/B2/B3: | for transitions of type "b" in which 1/2/3 thresholds are crossed; |
| C1: | for transitions of type "c"; |
| D1/D2/D3: | for transitions of type "d" in which 1/2/3 thresholds are crossed. |

The remainder of this section is devoted to describing the procedure followed in each of these cases.

### 3.1. Case A1: Transitions of Type (01a), (12a) and (23a)

In this case, $(x_{U_i}(n), U_i)$ is given by the intersection of a line segment that joins $(n-1, f_{n-1})$ and $(n, f_n)$ with $y = U_i$. Notice that $n - x_{U_i}(n)$ represents the delay by which the water level is affected following the crossing of $U_i$ (and consequent activation of the pump). There are three sub-cases to consider:

A1 (i). $d_m \leq n - x_{U_i}(n) \leq d_M$      A1 (ii). $n - x_{U_i}(n) < d_m$      A1 (iii). $n - x_{U_i}(n) > d_M$.

Case A1 (i) requires no intervention, since the delay is within the admissible values. If A1 (ii) occurs, we shift the maximum towards the right, along the line segment that joins $(n-1, f_{n-1})$ and $(n, f_n)$, in order to comply with the minimum delay, that is, a delay of $d_m$. Notice that the water level at the new maximum will not exceed $U_{i+1}$. In fact, since

the maximum is attained at $(x_{U_i}(n) + d_m, d_m r_n + U_i)$, for the water level to reach the next threshold, we would have

$$d_m r_n + U_i \geq U_{i+1} \Leftrightarrow d_m r_n \geq 0.2 \Leftrightarrow r_n \geq \frac{0.2}{d_m} \approx 11.97 \,,$$

which is impossible in this framework. Finally, in case A1 (iii), we split the the line segment that joins $(n-1, f_{n-1})$ and $(n, f_n)$ into two line segments. If $x_{U_i}(n) - d_M > 0$ and $\frac{U_i - f_{n-1}}{x_{U_i}(n) - d_M} < 2$, the break point will be at $x = x_{U_i}(n)$ along the line that contains $(n-1, f_{n-1})$ and $(x_{U_i}(n) - d_M, U_i)$—since $x_{U_i}(n) - d_M$ sits $d_M$ to the left of $x_{U_i}(n)$, we thus ensure a delay of $d_M$; since $\frac{U_i - f_{n-1}}{x_{U_i}(n) - d_M} < 2$, the water level at $x_{U_i}(n)$ will be at most $U_i + 0.1$, and so $U_{i+1}$ is not reached. If $x_{U_i}(n) - d_M < 0$ or $\frac{U_i - f_{n-1}}{x_{U_i}(n) - d_M} > 2$, the first line segment leaves $(n-1, f_{n-1})$ with a slope of $2r_n$ and halts at $\left(\frac{U_i - f_{n-1}}{2r_n} + d_M, U_i + 2d_M(f_n - f_{n-1})\right)$—again, the delay of $d_M$ is ensured, and, since $r_n < 2$ in this framework, $U_i + 2d_M r_n < U_{i+1}$ is guaranteed as well. In either case, the second line segment is the one that joins the break point and $(n, f_n)$.

Figure 4 illustrates sub-cases A1 (ii) and A1 (iii). The blue dots represent actual data entries (and blue line segments represent the water level as given directly by the readings), and the red dots represent additional break points, as given by the rules adopted (and red line segments represent the corrected water level). The purple (respectively, light blue) dashed lines help to locate the upward (respectively, downward) thresholds. This color code will be adopted throughout the entire work.

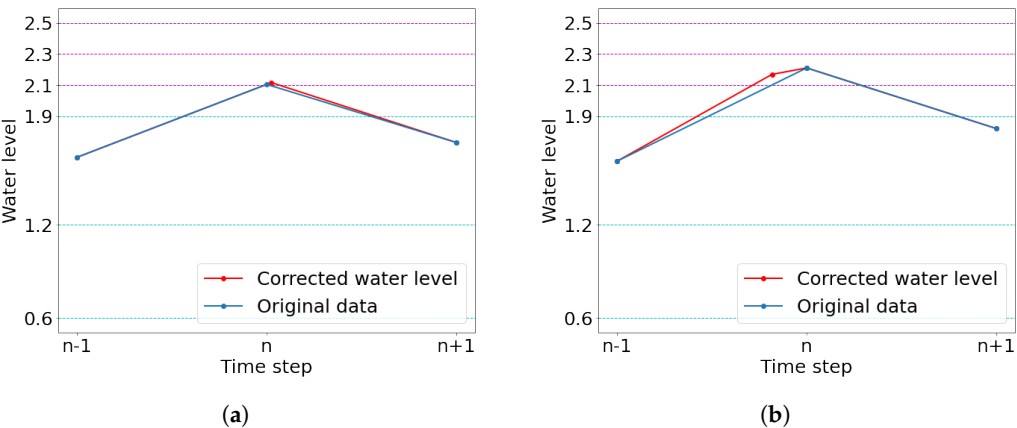

(**a**)                      (**b**)

**Figure 4.** Two sub-cases of case A1. (**a**) Illustration of sub-case A1 (ii): $f_{n-1} = 1.635$, $f_n = 2.104$ and $f_{n+1} = 1.73$. (**b**) Illustration of sub-case A1 (iii): $f_{n-1} = 1.61$, $f_n = 2.21$ and $f_{n+1} = 1.82$.

### 3.2. Case B1: Transitions of Type (01b), (12b) and (23b)

This turn, $(x_{U_i}(n), U_i)$ is given by the intersection of a line segment that joins $(n-1, f_{n-1})$ and $(x_I(n), f(x_I(n)))$ with $y = U_i$. We consider the following sub-cases:

B1 (i): $d_m \leq x_I(n) - x_{U_i}(n) \leq d_M$
B1 (ii): $x_I(n) - x_{U_i}(n) < d_m$
B1 (iii): $x_I(n) - x_{U_i}(n) > d_M$
B1 (iv): $f(x_I(n)) < U_i$.

Again, case B1 (i) requires no correction. In case B1 (ii), we proceed as in case A1 (ii), using $(x_I(n), f(x_I(n)))$ instead of $(n, f_n)$. In case B1 (iii), we lower $(x_I(n), f(x_I(n)))$ to $(x_I(n), \widetilde{y}_I)$, so that the line segment that joins $(n-1, f_{n-1})$ and $(x_I(n), \widetilde{y}_I)$ intersects $y = U_i$ in such a way that the delay equals $d_M$. To be precise, we have

$$\widetilde{y}_I = f_{n-1} + \frac{x_I(n)(U_i - f_{n-1})}{x_I(n) - d_M} \,.$$

Finally, in case B1 (iv), we proceed symmetrically to case B1 (iii), shifting the point $(x_I(n), f(x_I(n)))$ upwards to $(x_I(n), \widetilde{y}_I)$, but this turn occurs in order to get a delay of $d_m$, namely,

$$\widetilde{y}_I = U_i + \frac{U_i - f_{n-1}}{x_I(n) - d_m - (n-1)} \, .$$

Figure 5 illustrates sub-cases of (iii) and (iv) of case B1. The dotted line segments indicate the extension of the previous or future line segments, and so $(x_I(n), f(x_I(n)))$ is their intersection point.

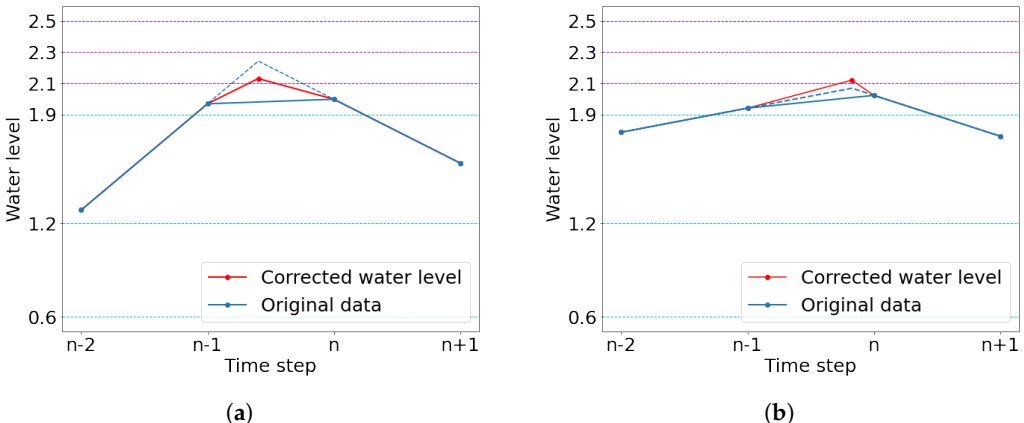

(a)                                   (b)

**Figure 5.** Two sub-cases of case B1. (**a**) Illustration of sub-case B1 (iii): $f_{n-1} = 1.968$ and $f_n = 1.997$ (with $f_{n-2} = 1.286$ and $f_{n+1} = 1.587$). (**b**) Illustration of sub-case B1 (iv): $f_{n-1} = 1.94$ and $f_n = 2.01$ (with $f_{n-2} = 1.785$ and $f_{n+1} = 1.76$).

### 3.3. Case C1: Transitions of Type (01c), (12c) and (23c)

In the case when $x_I(n)$ exists, but case B1 does not apply (and so $f(x_I(n)) < U_i - 0.05$) and the evidence of a threshold crossing comes from the sharp change from $r_{n-1}$ to $r_{n+1}$ (in this case, from a positive $r_{n-1}$ to a negative $r_{n+1}$) along with $f(x_I(n)) > f(x_I(n+1))$ (or the non-existence of $f(x_I(n+1))$), we nonetheless proceed as in case B1 (iv).

### 3.4. Case D1: Transitions of Type (01d), (12d) and (23d)

This turn, the evidence of a threshold crossing is given by a reading close to a threshold together with a significant change and opposite signs in $r_n$ to $r_{n+1}$. We proceed similarly in the two sub-cases

$$\text{D1 (i). } |r_n| > |r_{n+1}| \qquad\qquad \text{D1 (ii). } |r_n| \le |r_{n+1}|.$$

considering we extend the line segment with the greatest incline, allowing for a minimum delay. Figure 6 illustrates the sub-cases of (i) and (ii) of case D1, respectively.

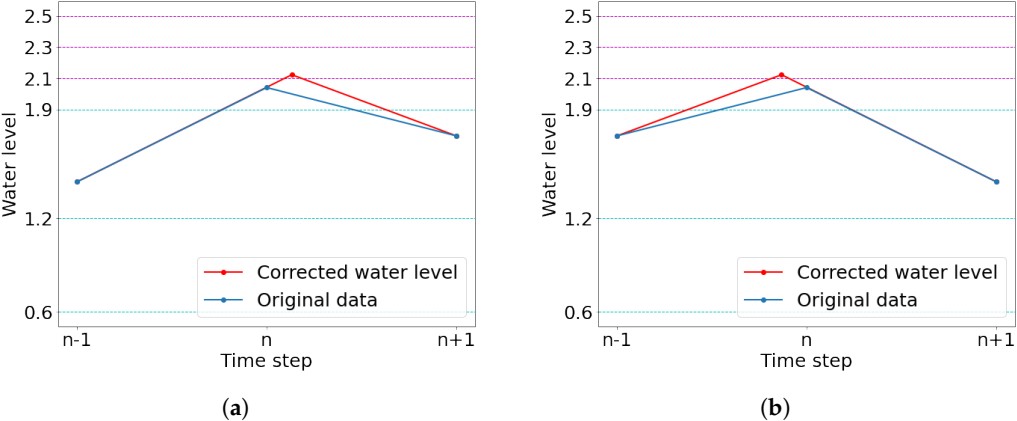

(a)                                   (b)

**Figure 6.** The 2 sub-cases of case D1. (**a**) Illustration of case D1 (i): $f_{n-1} = 1.435$, $f_n = 2.04$ and $f_{n+1} = 1.73$). (**b**) Illustration of case D1 (ii): $f_{n-1} = 1.73$, $f_n = 2.04$ and $f_{n+1} = 1.435$.

### 3.5. Case A2: Transitions of Type (02a) and (13a)

In this case, one or two additional break points are defined, following the same principles that were adopted in case A1. Given that the uncorrected version yields an exaggerated delay after the first threshold has been reached, we first proceed as in case A1 (iii). Therefore, we split the line segment between $(n-1, f_{n-1})$ and $(n, f_n)$ into two line segments, with a break point sitting $d_M$ to the left of the point where the line segment that joins $(n-1, f_{n-1})$ and $(n, f_n)$ meets the first threshold. Next, we consider the line segment that joins the first break point and $(n, f_n)$ and again apply the appropriate the sub-case of A1. Figure 7 illustrates an instance of case A2.

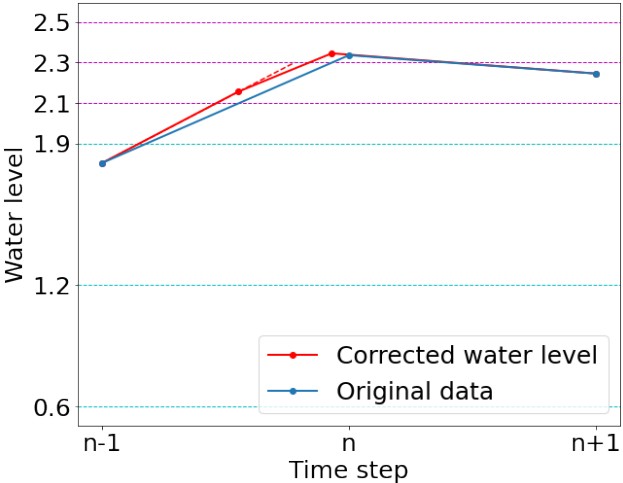

**Figure 7.** Illustration of case A2: $f_{n-1} = 1.804$, $f_n = 2.336$ and $f_{n+1} = 2.245$. The dotted extension of the first line segment shows how the slope of the water level decreases after the first break point. In this example, the line segment joining the first break point and $(n, f_n)$ yields an excessive delay, and so the second break point is again found as in A1 (iii).

### 3.6. Case B2: Transitions of Type (02b) and (13b)

As in case B1, $(x_{U_i}(n), U_i)$ is given by the intersection of a line segment that joins $(n-1, f_{n-1})$ and $(x_I(n), f(x_I(n)))$ with $y = U_i$. We set a first break point at $x = x_{U_i}(n)$ along the line that contains $(n-1, f_{n-1})$ and $(x_{U_i}(n) - d_M, U_i)$. In case the line segment that joins the first break point and $(x_I(n), f(x_I(n)))$ yields an admissible delay when crossing the second threshold, we take $(x_I(n), f(x_I(n)))$ to be the local maximum; otherwise, we proceed as in cases B1 (ii), (iii) or (iv), adjusting the maximum value $\tilde{y}_I$ in order of the corresponding delay to be within the interval $[d_m, d_M]$. Figures 8 illustrates case B2.

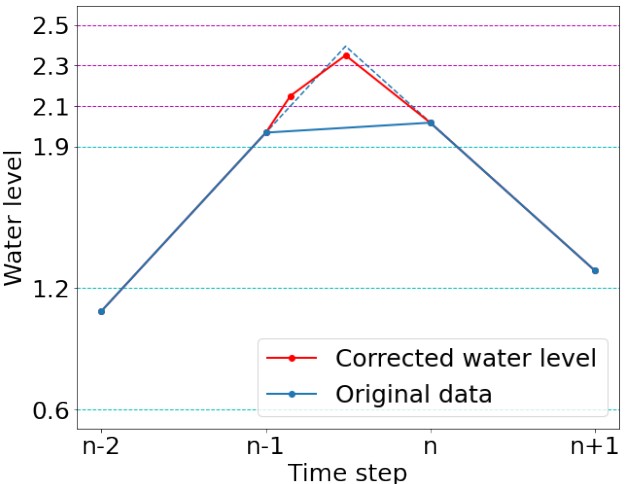

**Figure 8.** Illustration of case B2: $f_{n-1} = 1.968$ and $f_n = 2.017$ ( with $f_{n-2} = 1.086$ and $f_{n+1} = 1.287$).

### *3.7. Case D2: Transitions of Type (02d) and (13d)*

Here, we determine a first break point using the procedure established for case A1 (iii) and then apply the appropriate sub-case of case D1, according to whose line segment has the greatest incline: the line segment joining the first break point and $(n, f_n)$ or the line segment joining $(n, f_n)$ and $(n + 1, f_{n+1})$.

### *3.8. Case A3: Transitions of Type (03a)*

This case is dealt with analogously to case A2, now with the introduction of an extra initial break point. In order to obtain a more natural decay in the rate of the water level, we consider that the water level reaches $U_1$ at $x_{U_1}(n) - 2d_M$ (thus anticipating the crossing of the first threshold) and that the first break point occurs at $x_{U_1}(n) - d_M$.

### *3.9. Case B3: Transitions of Type (03b)*

Analogous to case A3, we introduce the extra initial break point, as defined there, and then proceed as in case B2.

### *3.10. Case D3: Transitions of Type (03d)*

Again, we introduce the extra initial break point and then proceed as in case D2.

## 4. Implementation on a Sample of the Data Set

In this section, we apply the algorithms from Sections 2 and 3 to a sample of the data set and analyse the results obtained. For purposes of this exercise, and since the initial reading is 0.699375, we consider that the status for the first reading, as given in the data set, is correct. Thus, $s(0) = 0$.

We begin by inferring the status of the system as described in Tables 3–6. The status change condition, if any, that applies to each row of the sample displayed in Table 2, together with the corrected status of the system, is indicated in Table 7. Figure 9 shows the water level as given by the data along with the points $(x_I(n), f(x_I(n)))$, when they exist. As we saw in Section 2, these points are instrumental when deciding the status transitions.

**Table 7.** Sequence of 20 rows of the data set with the system status correction.

| Time | Reading | Water Level | Original Status | Status Change Condition | Corrected Status |
|------|---------|-------------|-----------------|-------------------------|------------------|
| 12:10 | 0 | 0.699375 | 0 | — | 0 |
| 12:15 | 1 | 1.0975 | 0 | — | 0 |
| 12:20 | 2 | 2.1115625 | 1 | (01a) | 1 |
| 12:25 | 3 | 2.338125 | 1 | (12a) | 2 |
| 12:30 | 4 | 2.2928125 | 1 | (23b) | 3 |
| 12:35 | 5 | 1.9346875 | 1 | — | 3 |
| 12:40 | 6 | 1.445 | 1 | (32a) | 2 |
| 12:45 | 7 | 0.99875 | 1 | (21a) | 1 |
| 12:50 | 8 | 1.2565625 | 1 | — | 1 |
| 12:55 | 9 | 1.4234375 | 1 | — | 1 |
| 13:00 | 10 | 1.2134375 | 0 | — | 1 |
| 13:05 | 11 | 0.8346875 | 2 | — | 1 |
| 13:10 | 12 | 0.7 | 2 | — | 1 |
| 13:15 | 13 | 0.9665625 | 2 | (10b) | 0 |
| 13:20 | 14 | 2.51125 | 2 | (03a) | 3 |
| 13:25 | 15 | 1.955625 | 1 | — | 3 |
| 13:30 | 16 | 1.6290625 | 1 | (32a) | 2 |
| 13:35 | 17 | 1.2696875 | 1 | — | 2 |
| 13:40 | 18 | 1.576875 | 2 | (21d) | 1 |
| 13:45 | 19 | 1.82 | 1 | — | 1 |

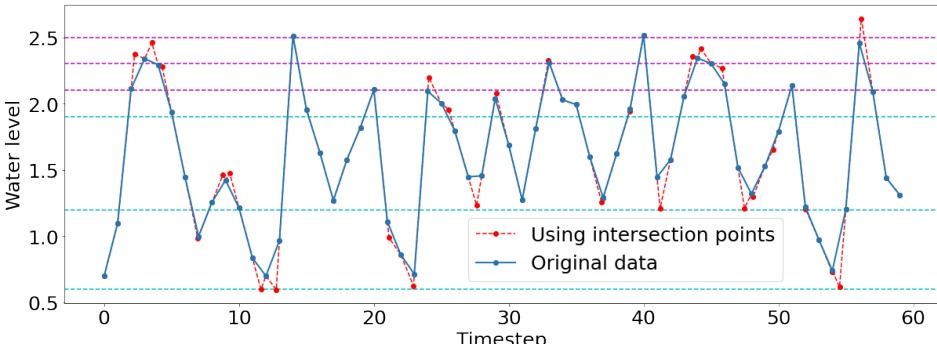

**Figure 9.** The water level using the readings and the existing intersection points $(x_I(n), f(x_I(n)))$ for a sample of the data set.

Figure 10 exhibits the comparison between the system status given in the data set and the corrected status at each time step (in black, as given by the data set, and in red, as given by the status inference set up in Section 2). Overall, this sample of the original data set yields a total of 78 5-min "active intervals", that is, 5-min intervals (some of which coincide in time) during which a pump was active, whereas the corrected version yields a total of 89 such intervals.

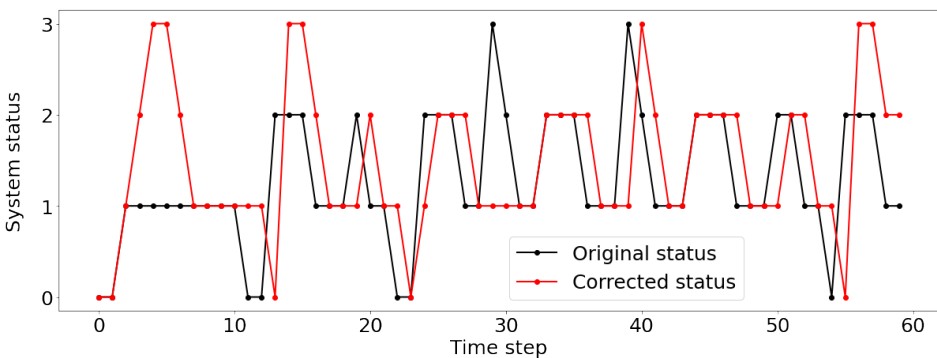

**Figure 10.** The system status for a sample of the data set, as given in the data set and by its correction using the rules established in Section 2.

For comparison, we remark that, in the original (complete) data set, there were 7971 5-min intervals in which no pump was working, 53,107 5-min intervals in which one pump was working, 8924 5-min intervals in which two pumps were working and 41 5-min intervals in which all pumps were working. Applying the status inference algorithm to the whole data set, these figures become 9251, 39,531, 18,120 and 3141, respectively. It is particularly striking how the full operation of the pumps was being underestimated.

Detailing individually for each of the three pumps, the correction of the status determined by Section 2 yields a total of 30 5-min active intervals for pump 1, 21 5-min active intervals for pump 2 and 38 5-min active intervals for pump 3. (This analysis, impossible to perform on the original data on account of its incoherence, uses the *modus operandi* of the system: when activating a pump, the system chooses the one that has been switched off longest and similarly for deactivation.)

Taking into account the modeling of the water level set up in Section 3, which always anticipates or delays the change of status, these values become 27.664, 20.115 and 38.076, respectively. Thus, comparing the un-modeled with the modeled version, pump 1 works approximately 8% less of the time, pump 2 works approximately 4% less of the time, pump 3 works roughly the same time, and the system works approximately 4.5% less of the time (the total having become 85.855 5-min units). Figure 11 displays the comparison between the moment when each pump is activated and deactivated (in black, as given by the status

inference algorithm from Section 2, and in red, as given by the modeling established in Section 3).

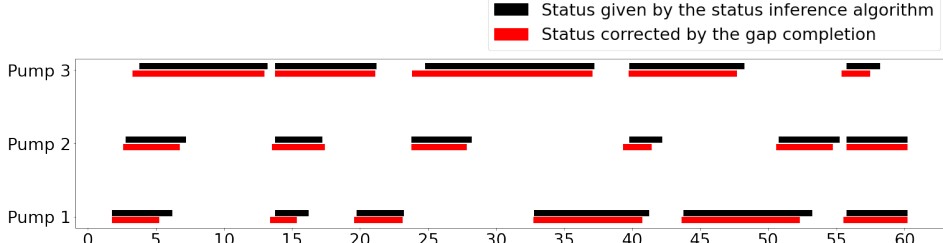

**Figure 11.** Timeline of the activation and deactivation of each pump for a sample of the data set, as given by the status inference algorithm from Section 2 and by its correction using the modeling established in Section 3.

Finally, Figure 12 displays the water level as given by the original data (blue dashed line) and the water level as modeled in Section 3 (red continuous line). For a better visualization, we focus on the first 20 entries of the sample under analysis. These graphs show how the gap completion preserves the water level readings and the extra break points introduced by this algorithm, with the consequent anticipation/retardation of the crossing of the thresholds when compared with the original data. The crossing of every appropriate threshold(s) and the delayed effect of the activation and deactivation of the pumps after each threshold crossing on the water level can also be observed. The outcome is that the response of the system is always coherent with its operating mode.

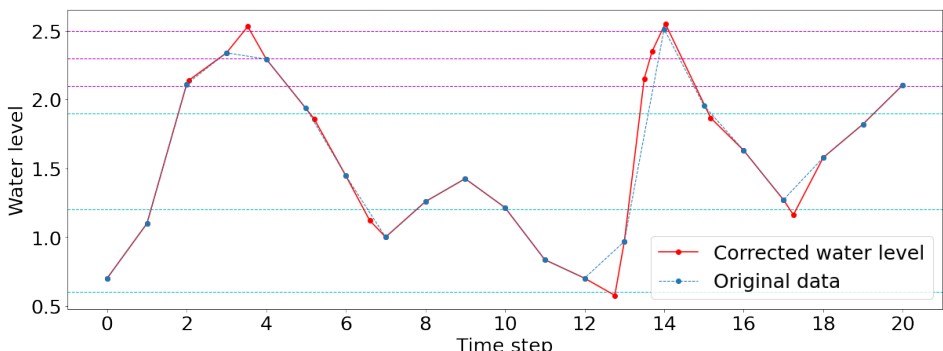

**Figure 12.** The water level for the first 20 entries of the sample, as given in the data set and by its correction using the modeling established in Section 3.

## 5. Conclusions

The algorithm defined in Section 2 allows for the extraction of the correct values of the status time series at every 5-min step, correcting the existing errors on the original data set. Although one of the objectives of this work, these figures alone do not provide the water management company the information needed on the number of active pumps on a continuous level. For example, if a pump was activated at 12:04:30 and deactivated one minute later, the status would be 1 for a time step, falsely implying that the pump worked for 5 min; if the same situation occurred at 12:03:00, the status would not reach the value 1 on the adjacent time steps, and it would be inferred that the pump had always been off. Both situations represent significant errors. With the algorithm described in Section 3, we were able to tackle this problem, providing a means to extract the desired information: the moment when each pump was activated or deactivated. As shown in Section 4, even for a small sample of the data set, the corrections, both on the status time series and on the amount, are not negligible.

As for future perspectives, this work could provide more accurate results if the data for the incoming water in the tank is taken into consideration. As of today, the water

management company only has daily data for this variable, which of course is not useful for the intended estimate. Another aspect that could be looked upon is the specificity of each pump. By evaluating the water lifting capacity of each pump, we could build a model that not only provides lifting profiles closer to reality, but also incorporates in the model possible differences between the pumps.

**Author Contributions:** Conceptualization, R.E. and F.S.; Methodology, R.E. and F.S.; Software, R.E. and F.S.; Validation, R.E. and F.S.; Formal analysis, R.E. and F.S.; Resources, R.E.; Writing—original draft, R.E. and F.S.; Writing—review & editing, R.E. and F.S. All authors have read and agreed to the published version of the manuscript.

**Funding:** This research received no external funding.

**Institutional Review Board Statement:** Not applicable.

**Informed Consent Statement:** Not applicable.

**Data Availability Statement:** The data set used in this work, as well as the code with the algorithm developed in *Python*, are available at the public GitHub repository: https://github.com/RicardoRoqueEng/DataSetGapCompletion.git (accessed on 15 February 2023).

**Acknowledgments:** The authors gratefully acknowledge the support given by FCT projects UIDB/04621/2020 and UIDP/04621/2020 of CEMAT at FC-Universidade de Lisboa.

**Conflicts of Interest:** The authors declare no conflict of interest.

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
