# Peer review of "Automatic Completion of Data Gaps Applied to a System of Water Pumps"

_mathematics, doi:10.3390/math11071707_

Round 1
Reviewer 1 Report
The paper presents the Automatic completion of data gaps applied to a water pumps system
-The paper deals with an interesting idea but needs major improvements.
As comments:
- The abstract must contain a sufficient summary of the work;
- Introduction missed the objective and the contribution of the paper,
- Authors should refer to the latest studies on the same subject and should make necessary comparisons.
- Authors have presented tables in the introductions. This is not suitable, so they must discuss them in the body of the manuscript,
- The applied method needs more explanations and details,
- Results and discussions are not sufficient; authors must revise their results,
- The paper without a conclusion, it must be included with perspectives,
- In this paper, authors are based on a limited number of references (4 refs), so the contribution must be compared and referenced with the latest and novel papers.
Author Response
Dear Reviewer,
First of all, we would like to thank you for your careful review of our work. We took into consideration your comments and suggestions and it seems to us that the work was greatly improved by them. We hope that the revised version meets your expectations.
As recommended, the manuscript has been revised by a native English speaker. We provide the responses to your comments inline below. Whenever indicated, line numbers of course correspond to the new version.
- The abstract must contain a sufficient summary of the work;
The abstract was revised in order to summarize the work in more detail.
- Introduction missed the objective and the contribution of the paper,
We added a comprehensive overview of the objectives and contributions of the paper in lines 108-123.
- Authors should refer to the latest studies on the same subject and should make necessary comparisons.
A more detailed review of the literature, and the corresponding comparisons, can now be found in lines 72-107.
- Authors have presented tables in the introductions. This is not suitable, so they must discuss them in the body of the manuscript,
Table 1 is a mere summary of the thresholds' information, useful for the reader's easy reference. Table 2 is crucial to illustrate the errors contained in the data set, without which it would not be possible to conveniently describe the objectives of the work. For this reason, we considered pertinent that both should be kept in the Introduction.
- The applied method needs more explanations and details,
Sections 2 and 3 have been rewritten in order to provide more thorough explanations on both the algorithms (cf. lines 166-211 and 213-248).
- Results and discussions are not sufficient; authors must revise their results,
With the improvements in the problem description in the Introduction and the improvements made to Sections 2 and 3, we trust that the results provided in Section 4 are now clear. Nontheless, we have made some improvements on this section.
- The paper without a conclusion, it must be included with perspectives,
A Conclusion section was added.
- In this paper, authors are based on a limited number of references (4 refs), so the contribution must be compared and referenced with the latest and novel papers.
Several other references were added in the Introduction, with the corresponding comparisons with our work.
Reviewer 2 Report
For a water pumps system with 3 water pumps, the paper presents an automatic completion method for data gaps. The following problems should be addressed:
1.The figures should be redrawn. The font is too small.
2.The purpose and contributions of the paper are not clear.
Author Response
Dear Reviewer,
First of all, we would like to thank you for your revision of our work. We took into consideration your comments and suggestions and it seems to us that the work was greatly improved by them. We hope that the revised version meets your expectations.
The manuscript has been revised by a native English speaker and we provide the response to your comments inline. Whenever indicated, line numbers of course correspond to the new version.
For a water pumps system with 3 water pumps, the paper presents an automatic completion method for data gaps. The following problems should be addressed:
1.The figures should be redrawn. The font is too small.
All figures have been redrawn. Apart from increasing the size if the font, we removed the figures' title, since the information it provided was already contained, in more detail, in the figure's caption and thus contributed with nothing but noise.
2.The purpose and contributions of the paper are not clear.
We added a comprehensive overview of the objectives and purpose of the paper in the Introduction (cf. lines 108-123), made some improvements on Section 4 and added a new final section, Conclusions, where the work's contributions are made clearer.
Reviewer 3 Report
This paper presents a method for data gap completion in a water pump system. The significance of such problem should be clarified.
Table 1 needs some more clarification. Are the threshold tank level? If so, what is the unit? The activate and deactivate actions need to be explained.
In Table 2, the unit for water level should be given. Are the number of activated pumps based on the rules given in Table 1? Why are there more activated pumps at readings 11 and 12 when the level is low compared to readings 2 to 5? Some explanations should be given.
The texts in Figure 1 are too small to read.
It would be better if the authors can use Table 1 and Figure 2 to illustrate the data gap issues.
It is mentioned at the start of Section 2 that “we consider the water level time series to be accurate, and so any incoherence between the water level readings and the number of active pumps is due to an error in the later”. If so, why do you need to correct the level data in the results given in Section 3?
Author Response
Dear Reviewer,
First of all, we would like to thank you for your careful review of our work. We took into consideration your comments and suggestions and it seems to us that the work was significantly improved by them. We hope that the revised version meets your expectations.
As recommended, the manuscript has been revised by a native English speaker. The responses to your comments are provided inline below. Whenever indicated, line numbers of course correspond to the new version.
This paper presents a method for data gap completion in a water pump system. The significance of such problem should be clarified.
We added a comprehensive overview of the objectives and purpose of the paper in the Introduction (cf. lines 108-123) and a new final section, Conclusions, where the work's contributions, and its significance in the real life context from where it comes, are made clearer.
Table 1 needs some more clarification. Are the threshold tank level? If so, what is the unit? The activate and deactivate actions need to be explained.
Units have been added, making now clear that the thresholds correspond to the level of the water in the tank. The activate and deactivate actions are explained in line 17.
In Table 2, the unit for water level should be given. Are the number of activated pumps based on the rules given in Table 1? Why are there more activated pumps at readings 11 and 12 when the level is low compared to readings 2 to 5? Some explanations should be given.
Units have been added and a typo corrected on Table 2. As explained in lines 45-55, the values on the 4th column are not consistent with the rules by which the system operates, thus containing several errors. We give two different examples of such errors, but do not pretend to list them exhaustively. The reason why the table has more entries than one may feel necessary at this point is because we will make use of them later in the work (cf. Table 7).
The texts in Figure 1 are too small to read.
All figures have been redrawn. Apart from increasing the size if the font, we removed the figures' title, since the information it provided was already contained, in more detail, in the figure's caption and thus contributed with nothing but noise.
It would be better if the authors can use Table 1 and Figure 2 to illustrate the data gap issues.
We added a remark in lines 49-51 that, together with the rest of this paragraph, makes the issues clear.
It is mentioned at the start of Section 2 that “we consider the water level time series to be accurate, and so any incoherence between the water level readings and the number of active pumps is due to an error in the later”. If so, why do you need to correct the level data in the results given in Section 3?
The original manuscript was misleading, when it stated, in the first paragraph of Section 3, that "We detail the corrections to make...". By this we meant that we would interfere with the water level between readings (as opposed to simply considering the water level to be given by the piecewise function made up of the line segments that join consecutive readings), but fully understand that our point wasn't clear. The term "corrections" was replaced by "procedure" and a great of the explanations given in this section were rewritten.
Round 2
Reviewer 1 Report
The revision that has been done improves relatively the manuscript.
- Conclusion needs some future perspectives
Author Response
Dear referee,
we are thankful for your careful revision of the manuscript.
In this round of revision we added a paragraph at the end of the final section concerning future perspectives:
"As for future perspectives, this work could provide more accurate results if the data for the incoming water in the tank is taken into consideration. As of today, the water management company only has daily data for this variable, which of course is not useful for the intended estimate.
Another aspect that could be looked upon is the specificity of each pump. By evaluating the water lifting capacity of each pump, we could build a model that not only provided lifting profiles closer to reality, but also to incorporate in the model possible differences between the pumps."
Best regards.
Reviewer 3 Report
The authors have adequately addressed my comments and the revised manuscript can be accepted.
Author Response
Dear referee,
Thank you again for your careful revision of the manuscript. In this round of revisions we just added a paragraph concerning future perspectives at the end of the final section:
"As for future perspectives, this work could provide more accurate results if the data for the incoming water in the tank is taken into consideration. As of today, the water management company only has daily data for this variable, which of course is not useful for the intended estimate.
Another aspect that could be looked upon is the specificity of each pump. By evaluating the water lifting capacity of each pump, we could build a model that not only provided lifting profiles closer to reality, but also to incorporate in the model possible differences between the pumps."
Best regards.